# Wave Aberration Correction for an Unobscured Off-Axis Three-Mirror Astronomical Telescope Using an Aberration Field Compensation Mechanism

**Jinxin Wang [1,2], Xu He [1,\*], Xiaohui Zhang [1], Mingze Ma [1,2] and Zhirui Cao [1]**

[1]  Changchun Institute of Optics, Fine Mechanics and Physics, Chinese Academy of Sciences, Changchun 130033, China
[2]  University of Chinese Academy of Sciences, Beijing 100049, China
\*  Correspondence: hexu_ciomp@sina.com

**Abstract:** Studying how to use the coupling characteristics of net aberration fields induced by different perturbation parameters to realize the wave aberration compensation correction of perturbed telescopes is of great significance for the development of active optics. Based on nodal aberration theory, this paper studies the wave aberration compensation correction method of an unobscured off-axis three-mirror telescope. Specifically, first of all, it theoretically analyzes the coupling effects and compensation relationships of net aberration fields induced by different perturbation parameters of the telescope. Furthermore, it establishes wave aberration correction models with the secondary mirror as the compensator and the third mirror as the compensator for the telescope, respectively. In the end, it verifies the two compensation correction models by simulations. The results show that the tolerance of the secondary mirror compensation correction mode (SMCM) to the perturbation parameter threshold is significantly better than that of the third mirror compensation correction model (TMCM). When the introduced perturbation parameter threshold is small, the correction accuracy of the two models for the wave aberrations is equivalent, and both reach the order of $10^{-3}\lambda$ (RMS, $\lambda = 632.8$ nm). When the perturbation parameter threshold is increased, the correction accuracy of SMCM can still be maintained at the order of $10^{-3}\lambda$ but the correction accuracy of TMCM would decrease by an order of magnitude.

**Keywords:** astronomical telescope; active optics; nodal aberration theory; aberration compensation; misalignments; figure errors

## 1. Introduction

Owning to the aperture obscuration caused by secondary mirrors (SM) and their supporting structure, on-axis reflective astronomical telescopes [1,2] not only lose light energy but also reduce the modulation transfer function (MTF) value at medium and low frequencies, so that the observation performances of optical systems are limited. On the basis of the on-axis reflective structure, the optical path structure with a field of view (FOV) off-axis, pupil off-axis, or a combination of the two is proposed to solve this problem [3–5]. Off-axis astronomical telescopes have the characteristics that their stray light is easy to control, their scattering property is low, and filled the pupil for wavefront sensing; these characteristics cause them to have a small and sharp diffraction pattern and high spatial resolution [6]. Compared with on-axis astronomical telescopes, when off-axis astronomical telescopes are in orbit, due to their special optical path structure, the slight perturbation caused by factors such as gravity, thermal stress, and mechanical oscillation will affect their imaging quality more easily. This brings challenges for active optical systems to complete wavefront correction in orbit.

The correction process of telescope perturbation parameters (rigid-body alignment errors and figure errors) is an important link in the active optical system [7,8]. The correction

results of perturbation parameters can be divided into restoration correction and compensation correction, according to different ideas of wave aberration correction. Restoration correction is obtained according to the idea of restoring the telescope to the nominal design, which requires the active optical system to be able to adjust all optical elements to achieve wave aberration correction. Compensation correction is obtained according to the idea of mutual compensation of optical elements, which requires the active optical system to adjust some of the optical elements to achieve wave aberration correction. However, in practical engineering, if the idea of restoration correction is followed, it is necessary to equip all optical components with corresponding multi-degree-of-freedom platforms (position and orientation adjustment) and force actuators (mirror surface figure adjustment). This approach is relatively expensive for space-based optical systems that have strict weight and volume constraints, both in construction cost and the structural stability of maintaining the system [9]. In addition, the correction efficiency (real-time) of an active optical system would be influenced by too many adjustment devices [10]. Therefore, studying the corresponding compensation correction methods is necessary to reduce the complexity of the active optical system and improve the correction efficiency.

For pupil-offset off-axis telescopes with designed tilts and decenters, discussed in this paper [11,12], the influence of different optical elements on wave aberrations has a strong coupling effect, which makes aberration compensation of this type of optical system easier to achieve. To correct the wave aberration of telescopes, the corresponding adjustment values (perturbation parameters) of optical elements should be determined. At present, the methods for solving these adjustment values mainly consist of traditional numerical methods and analytical methods based on nodal aberration theory (NAT). The numerical methods [13–16] do not consider field dependence of net aberrations induced by perturbation parameters when constructing a solution model for a specific optical system, and cannot analyze the coupling effect between different aberration fields from an analytical point of view, which makes it difficult to reasonably use this coupling effect to construct an optimal compensation correction model. The analytical methods based on NAT [17–23] are devoted to analyzing aberration field characteristics of perturbed optical systems, and deriving the complex functional relationship between perturbation parameters and aberration fields, which is of great significance for analyzing coupling between different perturbation parameters and constructing the corresponding optimal compensation correction model. In recent years, research on using NAT to quantitatively calculate perturbation parameters, to realize wave aberration correction of optical systems, has been reported. For instance, based on the idea of system restoration, Sebag et al. [24]. formulated the corresponding perturbation parameters solution scheme for the Large Synoptic Survey Telescope (LSST) by using NAT. In addition, Gu et al. [25,26] quantitatively calculated the perturbation parameters of both an on-axis three-mirror and an unobscured off-axis two-mirror telescope by using NAT. In addition, Schiesser et al. [27] designed and aligned an all-spherical unobscured four-mirror image relay for an ultra-broadband subpetawatt laser by using NAT. Moreover, Zhang [28] and Wang [11] studied the solution method of perturbation parameters for unobscured off-axis three-mirror telescopes based on NAT. The main purpose of these works was to restore perturbed optical systems to nominal design to the maximum extent by analytical solution.

As for analytical compensation correction of perturbed optical systems, some existing research mainly focused on the compensation methods of a single optical element. For example, Ju et al. [29] discussed how to use SM lateral misalignments to compensate for primary mirror (PM) astigmatic figure errors in an off-axis two-mirror telescope. Zhang et al. [30] and Wen et al. [31] studied how to use SM lateral misalignments to compensate for third-mirror (TM) lateral misalignments and for PM astigmatic figure errors in off-axis three-mirror telescopes, respectively. In addition, the compensation correction methods proposed in these studies did not consider the axial misalignments [32], PM coma figure errors, and PM spherical aberration figure errors. In order to compensate and correct telescopes with multiple perturbed optical elements at the same time, considering the axial

misalignments, PM coma figure errors, and PM spherical aberration figure errors, some new work is conducted in this paper. The paper mainly includes the following aspects: Section 2 describes the optical design parameters of an unobscured off-axis three-mirror telescope and theoretically analyzes the coupling effects and compensation relationships of net aberration fields induced by different perturbation parameters of the system. Section 3 describes the construction process of the compensation correction models for the off-axis three-mirror telescope. In Section 4, the simulation verification experiments are carried out. We conclude in Section 5.

## 2. Principle

### 2.1. Optical Path Structure and Design Parameters of an Unobscured Off-Axis Three-Mirror Telescope

Taking an unobscured off-axis three-mirror anastigmatic (TMA) telescope, with an aperture stop (located on the PM) of 0.5 m and an F-number of 12 used as an example, this paper aims to study its compensation correction method. The telescope is designed for the verification of key technology of space-based optical surveys. This type of off-axis optical system can be obtained by shifting the entrance pupil of an on-axis parent system with the same design parameters (e.g., radius, conic constants) and larger aperture. The optical path structure of the off-axis TMA telescope and the pupil transformation relationship with its on-axis parent system are both shown in Figure 1. In addition, the SM and TM of the optical system introduce tilts and decenters in the design process for the purpose of maximizing the balance of off-axis FOV aberrations and increasing the effective FOV of the telescope. The specific optical design parameters are given in Appendix A. The effective FOV of the telescope is $1° \times 1°$ (the central FOV is offset by $-0.5°$ in the y direction), and the average RMS wavefront error of full FOV is only $0.0417 \lambda$ ($\lambda = 632.8$ nm). The spectral range of the telescope is 255–1000 nm. One of its operating bands at 632.8 nm is selected as the evaluation unit of image quality in this paper. Evidently, it is a diffraction-limited system with competent imaging performance.

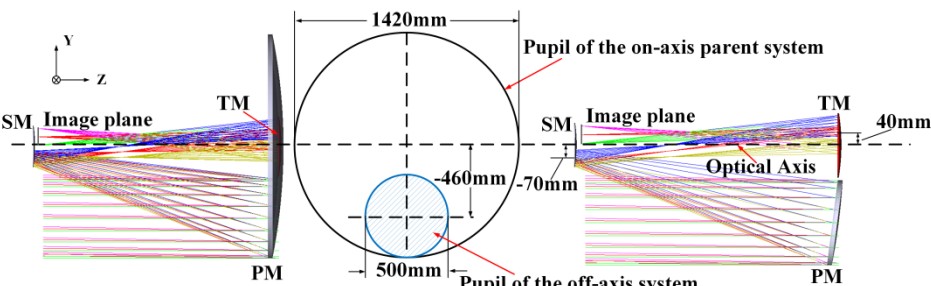

**Figure 1.** The optical path structure of the off-axis TMA telescope and pupil transformation relationship with its on-axis parent system.

### 2.2. Coupling Effect and Compensation Relationship of Net Aberration Fields Induced by Different Perturbation Parameters

According to NAT, when an optical system is perturbed, new aberrations will not be produced but the field dependence of aberrations (such as the linear dependence of the field, and the quadratic dependence of the field) will be changed [23,33]. However, different perturbation parameters often induce the same type of net aberration fields, that is, there are coupling effects among different perturbation parameters. They either compensate and offset each other to make the image quality better or accumulate and superimpose each other to make the image quality worse. This section aims to determine the compensation correction strategy for the off-axis TMA telescope described in the previous section by analyzing the coupling effect and compensation relationship between the net aberration fields induced by perturbation parameters. The wave aberration function of perturbed

pupil-offset off-axis optical systems with designed tilts and decenters can be expressed as [11]

$$
\left.
\begin{array}{c}
W_{off-axis} = \sum\limits_{j} \sum\limits_{p=0}^{\infty} \sum\limits_{n=0}^{\infty} \sum\limits_{m=0}^{\infty} \left(W_{klm} + \Delta W_{klm}\right)_j \left(\vec{H}_{Aj} \cdot \vec{H}_{Aj}\right)^p \left(\vec{\rho} \cdot \vec{\rho}\right)^n \left(\vec{H}_{Aj} \cdot \vec{\rho}\right)^m \\[4pt]
\vec{H}_{Aj} = \vec{H} - \vec{\sigma}^{\#}_j - \vec{\sigma}'_j \\[4pt]
\vec{\rho} = \frac{r}{R}\vec{\rho}' + \vec{h}
\end{array}
\right\} \quad (1)
$$

where $k = 2p + m$, $l = 2n + m$, $j$ is the optical surface number; $W_{klm}$ represents the aberration coefficient for a particular aberration type; $\Delta W_{klm}$ represents the net change of $W_{klm}$ induced by axial misalignments of optical surfaces; $\vec{\sigma}'$ denotes the aberration field decenter vector induced by lateral misalignments of optical surfaces; $\vec{\sigma}^{\#}$ denotes the aberration field decenter vector introduced in the design process of optical systems. Both $\vec{\sigma}'$ and $\vec{\sigma}^{\#}$ can be obtained by ray tracing using the method proposed in [34]. $\vec{H}_{Aj}$ and $\vec{H}$ represent the normalized effective field vectors before and after the aberration field center shift, respectively; $\vec{\rho}'$ and $\vec{\rho}$ represent the normalized pupil vectors of the off-axis system and its on-axis parent system, respectively; $\vec{h}$ represents the normalized position change vector of the pupil center of the off-axis system relative its on-axis parent system; $r$ and $R$ denote the pupil radius of the off-axis system and its on-axis parent system, respectively.

In our previous work [11], based on Equation (1), we derived the functions of third-order net astigmatic fringe Zernike coefficients (C5/C6), third-order net coma fringe Zernike coefficients (C7/C8), and third-order net spherical aberration fringe Zernike coefficient (C9) induced by axial and lateral misalignments. To facilitate the description of the proposed compensation correction principle, their field dependence is further clearly expressed in this section, as shown in the following equations.

$$
\left.
\begin{array}{l}
C_5 = \left[
\begin{array}{l}
M^2 \vec{h}_y \left(W_{131} + \Delta W_{131}\right)\vec{\sigma}'_{j,y} + \frac{M^2}{2}\left(W_{222} + \Delta W_{222}\right)\left(\vec{\sigma}'_j\vec{\sigma}'_j + 2\vec{\sigma}^{\#}_j\vec{\sigma}'_j\right)_x \\[6pt]
+ \frac{M^2}{2}\Delta W_{222}\left(\vec{\sigma}^{\#}_j\vec{\sigma}^{\#}_j\right)_x + M^2 \vec{h}_y\Delta W_{131}\vec{\sigma}^{\#}_{j,y} - 2M^2 \vec{h}_y^2\Delta W_{040}
\end{array}
\right]_C \\[18pt]
\quad + \left[
\begin{array}{l}
-M^2\vec{H}_x\left(W_{222} + \Delta W_{222}\right)\vec{\sigma}'_{j,x} + M^2\Delta W_{222}\left(\vec{H}_y\vec{\sigma}^{\#}_{j,y} - \vec{H}_x\vec{\sigma}^{\#}_{j,x}\right) \\[6pt]
-M^2\vec{h}_y\vec{H}_y\Delta W_{131} + M^2\vec{H}_y\left(W_{222} + \Delta W_{222}\right)\vec{\sigma}'_{j,y}
\end{array}
\right]_L \\[18pt]
\quad + \left[\frac{M^2}{2}\left(\vec{H}_x^2 - \vec{H}_y^2\right)\Delta W_{222}\right]_Q \\[14pt]
C_6 = \left[
\begin{array}{l}
-M^2\vec{h}_y\left(W_{131} + \Delta W_{131}\right)\vec{\sigma}'_{j,x} + \frac{M^2}{2}\left(W_{222} + \Delta W_{222}\right)\left(\vec{\sigma}'_j\vec{\sigma}'_j + 2\vec{\sigma}^{\#}_j\vec{\sigma}'_j\right)_y \\[6pt]
+ \frac{M^2}{2}\Delta W_{222}\left(\vec{\sigma}^{\#}_j\vec{\sigma}^{\#}_j\right)_y - M^2\vec{h}_y\Delta W_{131}\vec{\sigma}^{\#}_{j,x}
\end{array}
\right]_C \\[18pt]
\quad + \left[
\begin{array}{l}
-M^2\vec{H}_x\left(W_{222} + \Delta W_{222}\right)\vec{\sigma}'_{j,y} - M^2\Delta W_{222}\left(\vec{H}_y\vec{\sigma}^{\#}_{j,x} + \vec{H}_x\vec{\sigma}^{\#}_{j,y}\right) \\[6pt]
+ M^2\vec{h}_y\vec{H}_x\Delta W_{131} - M^2\vec{H}_y\left(W_{222} + \Delta W_{222}\right)\vec{\sigma}'_{j,x}
\end{array}
\right]_L \\[18pt]
\quad + \left[M^2\vec{H}_x\vec{H}_y\Delta W_{222}\right]_Q
\end{array}
\right\}, \quad (2)
$$

$$
\left.
\begin{array}{l}
C_7 = \left[-\frac{M^3}{3}\left(W_{131} + \Delta W_{131}\right)\vec{\sigma}'_{j,x} - \frac{M^3}{3}\Delta W_{131}\vec{\sigma}^{\#}_{j,x}\right]_C \\[12pt]
\quad + \left[\frac{M^3}{3}\vec{H}_x\Delta W_{131}\right]_L \\[12pt]
C_8 = \left[-\frac{M^3}{3}\left(W_{131} + \Delta W_{131}\right)\vec{\sigma}'_{j,y} - \frac{M^3}{3}\Delta W_{131}^d\vec{\sigma}^{\#}_{j,y} + \frac{4M^3}{3}\vec{h}_y\Delta W_{040}\right]_C \\[12pt]
\quad + \left[\frac{M^3}{3}\vec{H}_y\Delta W_{131}\right]_L
\end{array}
\right\}, \quad (3)
$$

$$C_9 = \left[\frac{M^4}{6}\Delta W_{040}\right]_C. \tag{4}$$

In Equations (2)–(4), the subscripts $C$, $L$, and $Q$ represent the field-constant dependence, field-linear dependence, and field-quadratic dependence of different aberration types, respectively; the subscripts $x$ and $y$ denote the x-component and y-components of a vector, respectively, $W_{klm} = \sum_j (W_{klm})_j$, $\Delta W_{klm} = \sum_j (\Delta W_{klm})_j$, and $M = \frac{r}{R}$. In addition, because the aperture of PM in astronomical telescopes is usually relatively large, it is prone to figure errors due to factors such as gravity, support stress, and thermal stress. In this paper, third-order astigmatic figure errors, third-order coma figure errors, and third-order spherical aberration figure errors of PM are specifically considered. When PM is used as the stop surface, the beams emitted from different FOVs entering the optical system have the same footprint on PM and completely cover PM. At that time, it can be considered that the aberration contribution of PM to different FOVs is the same, that is, the figure errors of PM will only introduce field-constant aberrations [22,26]. In consequence, we can add the field-constant aberrations induced by PM figure errors to the field-constant aberrations induced by misalignments. According to the relationship between optical path difference and figure error, Equations (2)–(4) can be modified as

$$\left.\begin{aligned} C_5 &= C_{5,C} + (n' - n)_{(FIGURE)}C_{5,C} + C_{5,L} + C_{5,Q} \\ C_6 &= C_{6,C} + (n' - n)_{(FIGURE)}C_{6,C} + C_{6,L} + C_{6,Q} \end{aligned}\right\}, \tag{5}$$

$$\left.\begin{aligned} C_7 &= C_{7,C} + (n' - n)_{(FIGURE)}C_{7,C} + C_{7,L} \\ C_8 &= C_{8,C} + (n' - n)_{(FIGURE)}C_{8,C} + C_{8,L} \end{aligned}\right\}, \tag{6}$$

$$C_9 = C_{9,C} + (n' - n)_{(FIGURE)}C_{9,C}, \tag{7}$$

where $n'$ and $n$, respectively, represent the refractive index of the space where the incident light and the reflected light are located (it could be noted that $n = 1$, $n' = -1$ for odd reflections and $n = -1$, $n' = 1$ for even reflections); $_{(FIGURE)}C_{5,C}$ and $_{(FIGURE)}C_{6,C}$ represent field-constant astigmatism induced by PM astigmatic figure errors; $_{(FIGURE)}C_{7,C}$ and $_{(FIGURE)}C_{8,C}$ represent field-constant coma induced by PM coma figure errors; $_{(FIGURE)}C_{9,C}$ represents the field-constant spherical aberration induced by PM spherical aberration figure errors.

In principle, only the same type of aberration fields can compensate for each other [33]. It can be seen from Equations (5)–(7) that misalignments and PM figure errors of the off-axis optical systems simultaneously introduce field-constant astigmatism, field-constant coma, and field-constant spherical aberration; this indicates that there are coupling effects between the aberration fields induced by misalignments and aberration fields induced by PM figure errors in the off-axis optical systems, and the possibility of mutual compensation exists. However, aberration compensation in the true sense includes not only the aberration fields induced by the target perturbation parameters (referring to PM figure errors) that need to be compensated for but also other types of aberration fields induced by compensation perturbation parameters (referring to misalignments), which need to be compensated at the same time. For instance, while using the field-constant aberrations induced by misalignments to compensate for the field-constant aberrations induced by PM figure errors, it can be seen from Equations (5)–(7) that misalignments will inevitably introduce additional field-linear aberrations and field-quadratic aberrations; these two types of aberration fields cannot be ignored in wide-field off-axis telescopes. To solve this problem, compensation perturbation parameters can be decomposed into different components, so that the different components can compensate for each other, thereby hopefully compensating for the extra-induced aberration fields. In fact, it is possible and relatively easy to decompose compensation perturbation parameters into different compensators according to different mirrors, that is, compensation between misalignments of different mirrors. This is mainly because misalignments of different mirrors will introduce the same

type of aberration fields, and no additional type of aberration fields will be introduced after compensation is done between different mirrors. Taking the off-axis three-mirror telescope as an example, when PM is used as the coordinate reference, the compensation parameters can be decomposed into two components, one of which is the misalignments of SM and the other is the misalignments of TM. The misalignments of SM will introduce field-constant aberrations, field-linear aberrations, and field-quadratic aberrations, and so does the misalignments of TM. Among them, field-constant aberrations induced by the misalignments of SM and TM can be used to compensate for field-constant aberrations induced by PM figure errors, while the additional field-linear aberrations and field-quadratic aberrations induced by the misalignments of SM and TM can compensate each other. Based on this analysis, we propose the following mechanism of aberration field compensation for an off-axis three-mirror telescope.

Taking SM to compensate misalignments of TM and figure errors of PM as an example, we can adjust SM to introduce appropriate field-constant aberrations (astigmatism, coma, spherical aberration), field-linear aberrations (astigmatism, coma) and field-quadratic aberrations (astigmatism). Among them, the introduced field-constant aberrations can be used to compensate for the field-constant aberrations caused by TM misalignments of PM figure errors. Meanwhile, the introduced field-linear aberrations and field-quadratic aberrations can be used to compensate for the field-linear aberrations and field-quadratic aberrations caused by TM misalignments.

In the following two sections, based on the above aberration field compensation mechanism, the optical compensation of misaligned TM and deformed PM by adjusting SM will be discussed; moreover, the optical compensation of misaligned SM and deformed PM by adjusting TM will also be discussed.

## 3. Compensation Correction Models for the Off-Axis TMA Telescope

To start with, we take SM as the compensator as an example to construct its corresponding compensation correction model, and the target objects to be compensated are TM misalignments and PM figure errors. Here, PM is chosen as the coordinate reference. To this end, Equations (5)–(7) can be modified as

$$\left.\begin{array}{l} C_{5,C}^{SM} + C_{5,L}^{SM} + C_{5,Q}^{SM} = C_5 - C_{5,C}^{TM} - C_{5,L}^{TM} - C_{5,Q}^{TM} - (n'-n)_{(FIGURE)}C_{5,C} \\ C_{6,C}^{SM} + C_{6,L}^{SM} + C_{6,Q}^{SM} = C_6 - C_{6,C}^{TM} - C_{6,L}^{TM} - C_{6,Q}^{TM} - (n'-n)_{(FIGURE)}C_{6,C} \end{array}\right\}, \tag{8}$$

$$\left.\begin{array}{l} C_{7,C}^{SM} + C_{7,L}^{SM} = C_7 - C_{7,C}^{TM} - C_{7,L}^{TM} - (n'-n)_{(FIGURE)}C_{7,C} \\ C_{8,C}^{SM} + C_{8,L}^{SM} = C_8 - C_{8,C}^{TM} - C_{8,L}^{TM} - (n'-n)_{(FIGURE)}C_{8,C} \end{array}\right\}, \tag{9}$$

$$C_{9,C}^{SM} = C_9 - C_{9,C}^{TM} - (n'-n)_{(FIGURE)}C_{9,C}. \tag{10}$$

In Equations (8) and (9), the terms containing superscripts *SM* and *TM*, respectively, represent the contribution of SM misalignments and TM misalignments to the aberration fields, and their numerical values can be solved by Equations (2)–(4). After the system being corrected, the net aberration fields induced by perturbation parameters is zero, so $C_{5\sim9} = 0$ in Equations (8) and (9). We aim to calculate the compensation adjustments of SM when the misalignments of TM and figure errors of PM are known. To achieve that, Equations (8)–(10) need to satisfy the aberration field compensation mechanism described in the previous section, as shown in the following equations:

$$\left.\begin{array}{l} C_{5,C}^{SM} = -\left[ C_{5,C}^{TM} + (n'-n)_{(FIGURE)}C_{5,C} \right] \\ C_{5,L}^{SM} = -C_{5,L}^{TM} \\ C_{5,Q}^{SM} = -C_{5,Q}^{TM} \\ C_{6,C}^{SM} = -\left[ C_{6,C}^{TM} + (n'-n)_{(FIGURE)}C_{6,C} \right] \\ C_{6,L}^{SM} = -C_{6,L}^{TM} \\ C_{6,Q}^{SM} = -C_{6,Q}^{TM} \end{array}\right\}, \tag{11}$$

$$C_{7,C}^{SM} = -\left[C_{7,C}^{TM} + (n' - n)_{(FIGURE)}C_{7,C}\right]$$
$$C_{7,L}^{SM} = -C_{7,L}^{TM}$$
$$C_{8,C}^{SM} = -\left[C_{8,C}^{TM} + (n' - n)_{(FIGURE)}C_{8,C}\right]$$
$$C_{8,L}^{SM} = -C_{8,L}^{TM}$$
$$\tag{12}$$

$$C_{9,C}^{SM} = -\left[C_{9,C}^{TM} + (n' - n)_{(FIGURE)}C_{9,C}\right]. \tag{13}$$

Based on Equations (11)–(13), some aberration coefficients of the $q$-th FOV points related to the compensation adjustments of SM can be obtained (hereinafter referred to as correlation coefficients), as shown in the following equations:

$$C_{5,C}^{SM} = f_1^{(q)}(x_{SM}, y_{SM}, \theta_{SM}, \phi_{SM}, z_{SM}), \tag{14}$$

$$C_{5,L}^{SM} = f_2^{(q)}(x_{SM}, y_{SM}, \theta_{SM}, \phi_{SM}, z_{SM}), \tag{15}$$

$$C_{5,Q}^{SM} = f_3^{(q)}(x_{SM}, y_{SM}, \theta_{SM}, \phi_{SM}, z_{SM}), \tag{16}$$

$$C_{6,C}^{SM} = f_4^{(q)}(x_{SM}, y_{SM}, \theta_{SM}, \phi_{SM}, z_{SM}), \tag{17}$$

$$C_{6,L}^{SM} = f_5^{(q)}(x_{SM}, y_{SM}, \theta_{SM}, \phi_{SM}, z_{SM}), \tag{18}$$

$$C_{6,Q}^{SM} = f_6^{(q)}(x_{SM}, y_{SM}, \theta_{SM}, \phi_{SM}, z_{SM}), \tag{19}$$

$$C_{7,C}^{SM} = f_7^{(q)}(x_{SM}, y_{SM}, \theta_{SM}, \phi_{SM}, z_{SM}), \tag{20}$$

$$C_{7,L}^{SM} = f_8^{(q)}(x_{SM}, y_{SM}, \theta_{SM}, \phi_{SM}, z_{SM}), \tag{21}$$

$$C_{8,C}^{SM} = f_9^{(q)}(x_{SM}, y_{SM}, \theta_{SM}, \phi_{SM}, z_{SM}), \tag{22}$$

$$C_{8,L}^{SM} = f_{10}^{(q)}(x_{SM}, y_{SM}, \theta_{SM}, \phi_{SM}, z_{SM}), \tag{23}$$

$$C_{9,C}^{SM} = f_{11}^{(q)}(x_{SM}, y_{SM}, \theta_{SM}, \phi_{SM}, z_{SM}). \tag{24}$$

In Equations (14)–(24), $x_{SM}$ and $y_{SM}$ represent the linear compensation adjustments of SM in the x- and y-axis, respectively; $\theta_{SM}$ and $\phi_{SM}$ represent the angular compensation adjustments of SM about the x- and y-axis, respectively; $z_{SM}$ represent the axial compensation adjustments of SM in z-axis; $f_i^{(q)}(i = 1, 2 \cdots, 11)$ represents the functional relationship between the correlation coefficients and the compensation adjustments of SM, which is a complex nonlinear equation system. Considering that the values of misalignments in active optics are very small (Generally, linear misalignments are on the level of micrometers, and angular misalignments are on the level of arcseconds [30]). Therefore, using the power series expansion and selecting only the first-order term of the power series, $f_i^{(q)}(i = 1, 2 \cdots, 11)$ can be approximately replaced by a linear system of equations:

$$f_i^{(q)} = f_{0i}^{(q)} + \frac{\partial f_i^{(q)}}{\partial(x_{SM})}x_{SM} + \frac{\partial f_i^{(q)}}{\partial(y_{SM})}y_{SM} + \frac{\partial f_i^{(q)}}{\partial(\theta_{SM})}\theta_{SM} + \frac{\partial f_i^{(q)}}{\partial(\phi_{SM})}\phi_{SM} + \frac{\partial f_i^{(q)}}{\partial(z_{SM})}z_{SM}, \tag{25}$$

where $f_{0i}^{(q)}$ is a constant related to the structural parameters and Gaussian parameters of the system; $\frac{\partial f_i^{(q)}}{\partial()}$ represents the first-order partial derivative of the correlation coefficients to the adjustments of each dimension of SM. By selecting some FOV points that can cover the full FOV, the overdetermined linear equation system formed by Equation (25) can be used to calculate the compensation adjustments of SM. It should be noted that $f_{0i}^{(q)}$ and

$\frac{\partial f_i^{(q)}}{\partial()}$ are obtained by fitting based on the principle of multiple linear regression [11]. The specific fitting method is as follows:

1.  Making each dimension of SM randomly generate N groups of perturbations according to a standard uniform distribution within a certain range. Meanwhile, bringing the generated N groups of perturbations into Equations (2)–(4), so that N groups of sample data can be obtained:

$$\left( x_{SM,t}, y_{SM,t}, \theta_{SM,t}, \phi_{SM,t}, z_{SM,t}; f_{i,t}^{(q)} \right) = t = 1, 2, \cdots, N \tag{26}$$

2.  Integrating this batch of sample data into the structural form shown in Equation (25), so the following equations could be obtained:

$$\left. \begin{aligned} f_{i,1}^{(q)} &= f_{0i} + \frac{\partial f_i^{(q)}}{\partial(x_{SM})} x_{SM,1} + \frac{\partial f_i^{(q)}}{\partial(y_{SM})} y_{SM,1} + \frac{\partial f_i^{(q)}}{\partial(\theta_{SM})} \theta_{SM,1} + \frac{\partial f_i^{(q)}}{\partial(\phi_{SM})} \phi_{SM,1} + \frac{\partial f_i^{(q)}}{\partial(z_{SM})} z_{SM,1} \\ f_{i,2}^{(q)} &= f_{0i} + \frac{\partial f_i^{(q)}}{\partial(x_{SM})} x_{SM,2} + \frac{\partial f_i^{(q)}}{\partial(y_{SM})} y_{SM,2} + \frac{\partial f_i^{(q)}}{\partial(\theta_{SM})} \theta_{SM,2} + \frac{\partial f_i^{(q)}}{\partial(\phi_{SM})} \phi_{SM,2} + \frac{\partial f_i^{(q)}}{\partial(z_{SM})} z_{SM,2} \\ &\vdots \\ f_{i,N}^{(q)} &= f_{0i} + \frac{\partial f_i^{(q)}}{\partial(x_{SM})} x_{SM,N} + \frac{\partial f_i^{(q)}}{\partial(y_{SM})} y_{SM,N} + \frac{\partial f_i^{(q)}}{\partial(\theta_{SM})} \theta_{SM,N} + \frac{\partial f_i^{(q)}}{\partial(\phi_{SM})} \phi_{SM,N} + \frac{\partial f_i^{(q)}}{\partial(z_{SM})} z_{SM,N} \end{aligned} \right\}. \tag{27}$$

Equation (27) is the mathematical model of the constructed multiple linear regression.

3.  Writing Equation (27) in the form of a matrix so as to get

$$\alpha = \beta \chi, \tag{28}$$

where

$$\alpha = \left( f_{i,1}^{(q)} \quad f_{i,2}^{(q)} \quad \cdots \quad f_{i,N}^{(q)} \right)^T, \tag{29}$$

$$\beta = \begin{pmatrix} 1 & x_{SM,1} & y_{SM,1} & \theta_{SM,1} & \phi_{SM,1} & z_{SM,1} \\ 1 & x_{SM,2} & y_{SM,2} & \theta_{SM,2} & \phi_{SM,2} & z_{SM,2} \\ \vdots & \vdots & \vdots & \vdots & \vdots & \vdots \\ 1 & x_{SM,N} & y_{SM,N} & \theta_{SM,N} & \phi_{SM,N} & z_{SM,N} \end{pmatrix}, \tag{30}$$

$$\chi = \left( f_{0i}^{(q)} \quad \frac{\partial f_i^{(q)}}{\partial(x_{SM})} \quad \frac{\partial f_i^{(q)}}{\partial(y_{SM})} \quad \frac{\partial f_i^{(q)}}{\partial(\theta_{SM})} \quad \frac{\partial f_i^{(q)}}{\partial(\phi_{SM})} \quad \frac{\partial f_i^{(q)}}{\partial(z_{SM})} \right)^T. \tag{31}$$

Finally, using the least squares method to estimate $\chi$, we can get

$$\chi = \left( \beta^T \beta \right)^{-1} \beta^T \alpha, \tag{32}$$

where the superscript $T$ represents the matrix transpose operation; the superscript $-1$ represents the matrix inversion operation.

In the above, a correction model for compensating TM misalignments and PM figure errors is constructed by using SM as the compensator. When using TM as the compensator to construct a correction model for compensating SM misalignments and PM figure errors, it is only necessary to modify Equations (8)–(10) as the following equations:

$$\left. \begin{aligned} C_{5,C}^{TM} + C_{5,L}^{TM} + C_{5,Q}^{TM} &= C_5 - C_{5,C}^{SM} - C_{5,L}^{SM} - C_{5,Q}^{SM} - (n' - n)_{(FIGURE)} C_{5,C} \\ C_{6,C}^{TM} + C_{6,L}^{TM} + C_{6,Q}^{TM} &= C_6 - C_{6,C}^{SM} - C_{6,L}^{SM} - C_{6,Q}^{SM} - (n' - n)_{(FIGURE)} C_{6,C} \end{aligned} \right\}, \tag{33}$$

$$\left. \begin{aligned} C_{7,C}^{TM} + C_{7,L}^{TM} &= C_7 - C_{7,C}^{SM} - C_{7,L}^{SM} - (n' - n)_{(FIGURE)} C_{7,C} \\ C_{8,C}^{TM} + C_{8,L}^{TM} &= C_8 - C_{8,C}^{SM} - C_{8,L}^{SM} - (n' - n)_{(FIGURE)} C_{8,C} \end{aligned} \right\}, \tag{34}$$

$$C_{9,C}^{TM} = C_9 - C_{9,C}^{SM} - (n' - n)_{(FIGURE)} C_{9,C}. \tag{35}$$

The calculation method of the compensation adjustments required for the TM is similar to the calculation method of the SM, and will not be repeated here.

## 4. Verification of Compensation Correction Models

### 4.1. A Specific Compensation Correction Case

In this section, the off-axis TMA telescope described in Section 2 will be used to verify the constructed compensation correction models. The specific steps of the simulation experiment using the SM as the compensator to compensate for TM misalignments and PM figure errors are as follows:

1.  In the optical design model of the off-axis three-mirror telescope (ZEMAX software is used for modeling in this paper), the perturbation parameters (TM misalignments and PM figure errors) shown in Table 1 are introduced randomly.
2.  The dynamic data linking function of MATLAB and ZEMAX is used to calculate the full-field distributions of the Fringe Zernike astigmatism(C5/6), the Fringe Zernike coma(C7/8), and the Fringe Zernike spherical aberration(C9) before and after perturbation of the telescope, they are calculated with $12 \times 12$ equally spaced FOV points in $1° \times 1°$ and shown in Figure 2a,b, Figure 3a,b, Figure 4a,b, respectively.
3.  By using the SM compensation correction model constructed in Section 3, the linear compensation adjustments, angle compensation adjustments, and axial compensation adjustments required by SM can be calculated, as shown in Table 2.
4.  The calculated compensation adjustments are introduced into the perturbed telescope and the full-field distributions of the Fringe Zernike astigmatism(C5/6), the Fringe Zernike coma(C7/8), and the Fringe Zernike spherical aberration(C9) after compensation adjustment of the telescope are calculated, as shown in Figures 2c, 3c and 4c.
5.  In addition, in order to evaluate the compensation correction accuracy more clearly, Figure 5a–c show the full-field distributions of C5/6, C7/8, and C9 differences before perturbation and after compensation adjustment of the telescope, respectively.

**Table 1.** The introduced figure errors of PM and misalignments of TM.

| $(FIGURE)^{C_{5,C}}$ | $(FIGURE)^{C_{6,C}}$ | $(FIGURE)^{C_{7,C}}$ | $(FIGURE)^{C_{8,C}}$ | $(FIGURE)^{C_{9,C}}$ |
|---|---|---|---|---|
| $-0.07\lambda$ | $0.08\lambda$ | $0.04\lambda$ | $0.06\lambda$ | $0.08\lambda$ |
| $x_{TM}$ | $y_{TM}$ | $\theta_{TM}$ | $\phi_{TM}$ | $z_{SM}$ |
| $-0.03$ mm | $0.07$ mm | $0.006°$ | $-0.004°$ | $0.05$ mm |

**Table 2.** The calculated compensation adjustments for SM.

| $x_{SM}$ | $y_{SM}$ | $\theta_{TM}$ | $\phi_{TM}$ | $z_{SM}$ |
|---|---|---|---|---|
| $0.1019$ mm | $0.0485$ mm | $-0.0012°$ | $0.0019°$ | $0.0233$ mm |

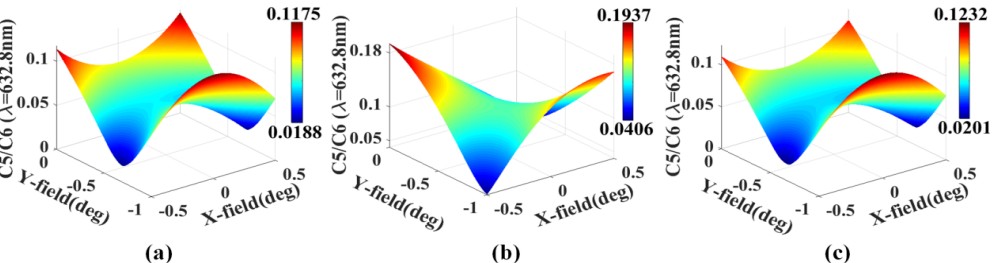

**Figure 2.** Distribution of Fringe Zernike astigmatism (C5/6) in the full-field of the off-axis TMA telescope: (**a**) before perturbation, (**b**) after perturbation, and (**c**) after compensation correction.

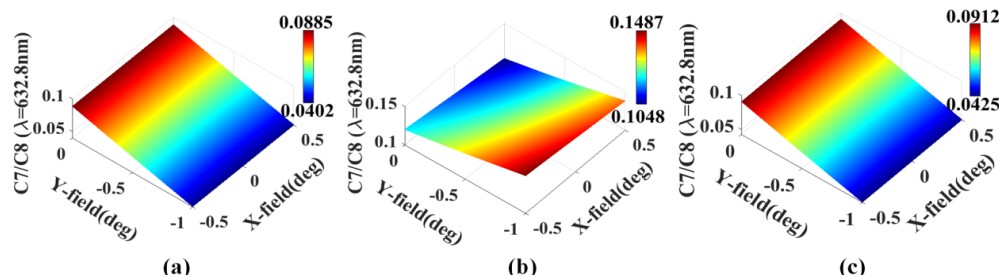

**Figure 3.** Distribution of Fringe Zernike coma (C7/8) in the full-field of the off-axis TMA telescope: (**a**) before perturbation, (**b**) after perturbation, and (**c**) after compensation correction.

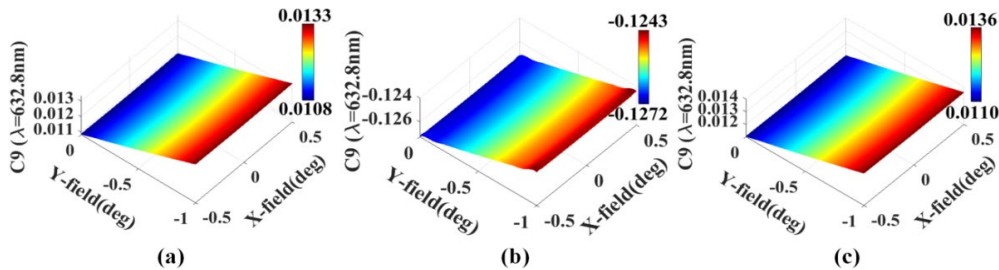

**Figure 4.** Distribution of Fringe Zernike spherical aberration (C9) in the full-field of the off-axis TMA telescope: (**a**) before perturbation, (**b**) after perturbation, and (**c**) after compensation correction.

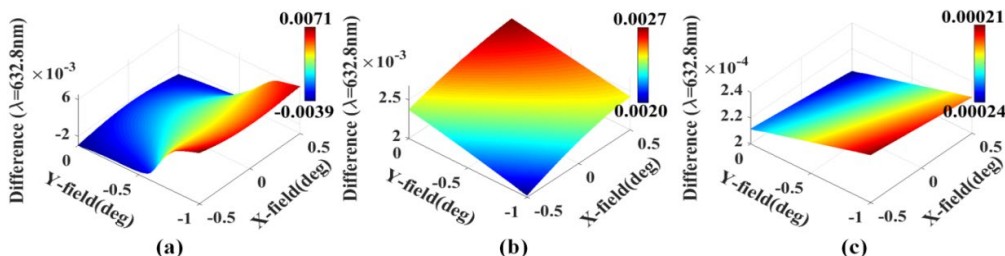

**Figure 5.** The full-field distribution of (**a**) C5/6, (**b**) C7/8, and (**c**) C9 differences before perturbation and after compensation adjustment of the off-axis TMA telescope.

By comparing Figure 2a and Figure 2b, Figure 3a and Figure 3b, and Figure 4a and Figure 4b, respectively, it can be seen that the full-field distribution of astigmatism, coma, and spherical aberration of the system changes obviously compared with the design state (it should be noted that the numerical scale of each sub-figure is different) after introducing the misalignments and figure errors. Specifically, the full-field distribution of astigmatism and coma changes not only in amplitude but also in shape; however, the change of the full-field distribution of spherical aberration is only reflected in the amplitude, and the shape of its full-field distribution does not change. As analyzed in Section 2, the main reason for this phenomenon is that the misalignments and PM astigmatic figure errors will introduce field-constant astigmatism, field-linear astigmatism, and field-quadratic astigmatism, and the coupling of these three types of aberration fields will change the shape of the full-field distribution of astigmatism. Likewise, the coupling effect of field-constant coma and field-linear coma induced by misalignments and PM coma figure errors causes the shape of the full-field distribution of the coma to change. However, the misalignments and PM spherical aberration figure errors only introduce field-constant spherical aberration, which makes the shape of the full field-of-view distribution of the spherical aberration unchanged.

By comparing Figure 2a and Figure 2c, Figure 3a and Figure 3c, and Figure 4a and Figure 4c, respectively, it can be seen that the full-field distribution of astigmatism, coma, and spherical aberration of the telescope after compensation correction are close to the design state in terms of both amplitude and shape. It can be seen from Figure 5 that the correction accuracy of astigmatism and coma of the telescope in the full field-of-view

reached the order of $10^{-3}$, and the correction accuracy of spherical aberration reached the order of $10^{-4}$. This indicates that the aberrations induced by TM misalignments and the PM figure errors are well compensated and corrected by adjusting SM.

When verifying the TM compensation correction model (TMCM), we introduced the same values of misalignments and figure shape errors in SM and PM of the telescope as shown in Table 1, and the results of compensation correction are similar to those of the SM compensation correction model (SMCM). Since this section verifies the correctness of different compensation correction models only under a specific perturbation state, this leads to the inability to comprehensively and objectively compare the correction capabilities of these two compensation correction models. Therefore, we will compare these two compensation correction models by Monte Carlo analyses in the next section.

*4.2. Monte Carlo Analyses and Comparison for Different Compensation Correction Models*

In this section, Monte Carlo analyses will be conducted on SMCM and TMCM within the perturbation ranges of the misalignments, figure errors are shown in Table 3. The simulation analyses are divided into four cases, including the first case, the second case, the third case, which has increases in turn in terms of the perturbation ranges, and the fourth case, which has the same perturbation range as the first but contains a 5% measurement error. For the Monte Carlo analyses of SMCM, within the given perturbation ranges, 500 sets of perturbation parameters (TM misalignments and PM figure errors) were randomly generated according to a standard uniform distribution, representing 500 random perturbation systems to be corrected by SMCM. For the Monte Carlo analyses of TMCM, also within the given perturbation ranges, 500 sets of perturbation parameters (SM misalignments and PM figure errors) were randomly generated according to the standard uniform distribution, representing 500 random perturbation systems to be corrected by TMCM. Here we choose the average RMS wavefront error (WFE) of the full FOV to evaluate the compensation correction effects of these two models (ignoring the piston, tilt, and defocus terms). After simulation, the compensation correction results of the SMCM and TMCM are shown in Figures 6 and 7, respectively. Furthermore, Table 4 shows the root-mean-square deviation of the average RMS WFE under different cases, and the definition of the root-mean-square deviation is shown in the following equation:

$$RMSD = \sqrt{\frac{1}{n}\sum_{i=1}^{n}\left[\left(WFE^{Average}\right)_i^{correction} - \left(WFE^{Average}\right)_i^{design}\right]^2} \qquad (36)$$

where $n$ is the number of perturbed samples ($n = 500$); $\left(WFE^{Average}\right)_i^{correction}$ is the average RMS WFE after compensation correction for perturbation sample $i$; $\left(WFE^{Average}\right)_i^{design}$ is the average RMS WFE in nominal design (0.0417 λ). Equation (36) can be used to evaluate the correction accuracy of the compensation correction models.

**Table 3.** Perturbation ranges of misalignments and figure errors.

|  | Linear Misalignment (mm) | Angular Misalignment (deg) | $_{(FIGURE)}C(\lambda)$ |
| --- | --- | --- | --- |
| Case 1 | [−0.1 0.1] | [−0.01 0.01] | [−0.05 0.05] |
| Case 2 | [−0.2 0.2] | [−0.02 0.02] | [−0.1 0.1] |
| Case 3 | [−0.3 0.3] | [−0.03 0.03] | [−0.15 0.15] |
| Case 4 | [−0.1 0.1] | [−0.01 0.01] | [−0.05 0.05] |
| With 5% measurement error | | | |

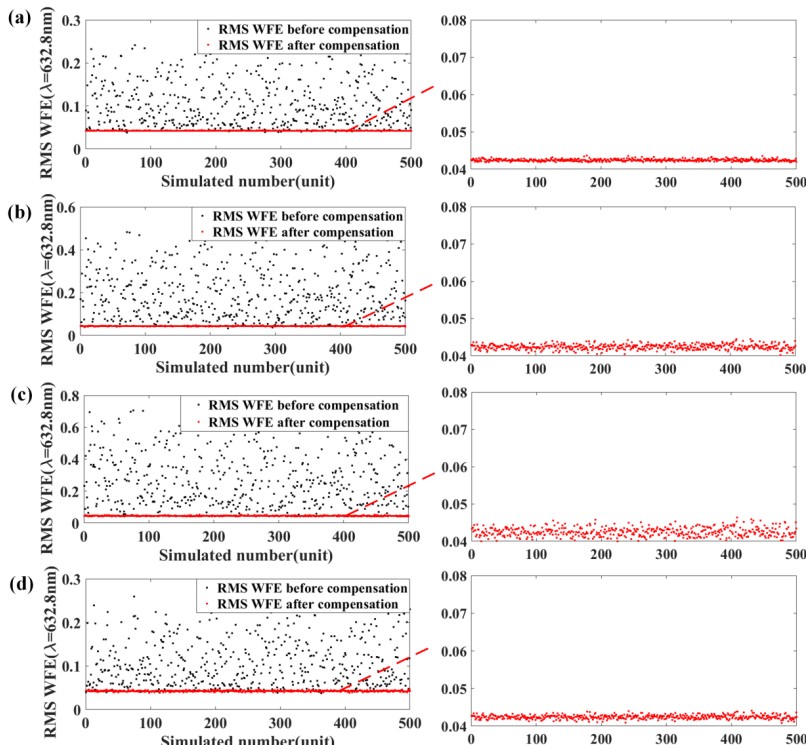

**Figure 6.** Average RMS WFE before and after compensation of the optical system based on SMCM: (**a**) Case 1, (**b**) Case 2, (**c**) Case 3, and (**d**) Case 4. It could be noted that the black and red spots represent the average RMS WFE before and after compensation, respectively.

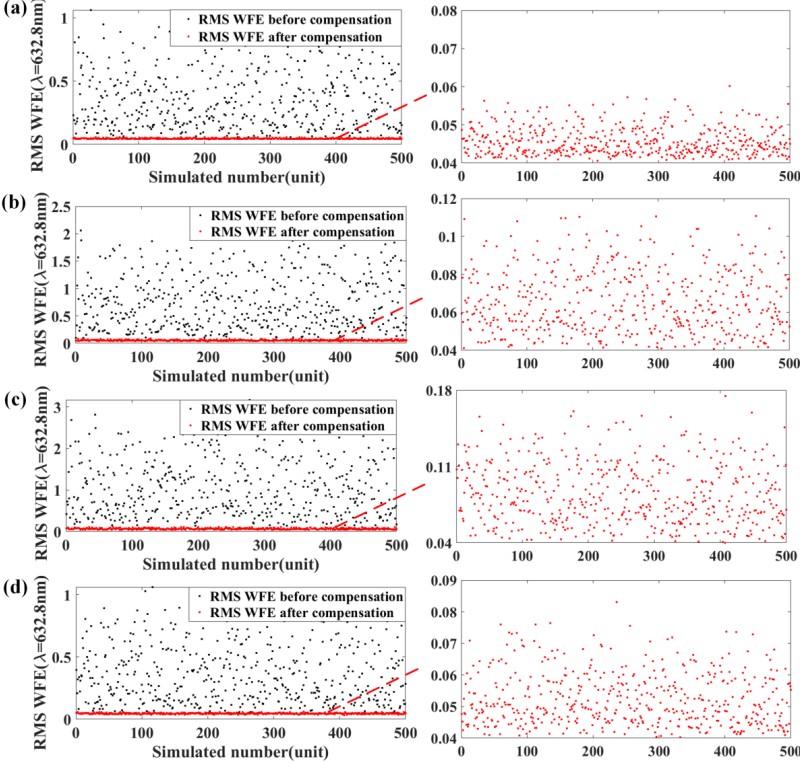

**Figure 7.** Average RMS WFE before and after compensation of the optical system based on TMCM: (**a**) Case 1, (**b**) Case 2, (**c**) Case 3, and (**d**) Case 4. It could be noted that the black and red spots represent the average RMS WFE before and after compensation, respectively.

**Table 4.** The RMSD of SMCM and TMCM in different cases.

|  | Case 1 | Case 2 | Case 3 | Case 4 |
|---|---|---|---|---|
| SMCM | $9.672 \times 10^{-4}$ | $2.221 \times 10^{-3}$ | $3.466 \times 10^{-3}$ | $1.741 \times 10^{-3}$ |
| TMCM | $7.400 \times 10^{-3}$ | $1.979 \times 10^{-2}$ | $4.024 \times 10^{-2}$ | $1.558 \times 10^{-2}$ |

As can be seen from Figure 6, SMCM can obtain convergent correction results for different cases. Anyway, it can be seen from Table 4 that the correction accuracy of the SMCM is basically maintained at the order of $10^{-3}$ under different cases. In addition, as can be seen from Figure 7—although TMCM also achieved convergent correction results—by comparing Figures 6 and 7, it can be found that the correction performance of TMCM is significantly worse than that of SMCM. This is also verified by the correction accuracy results of TMCM in Table 4, especially in Case 2, Case 3, and Case 4, the correction accuracy of TMCM decreases by one order of magnitude compared with that of SMCM. The reason for this phenomenon may be that in the off-axis TMA telescope, the wave aberrations of the system are more sensitive to the SM misalignments than the TM misalignments (it can be seen by comparing the numerical scale of the ordinate in Figures 6 and 7), that is to say, under the same perturbation range, the contribution of SM misalignments to the system aberrations is greater than the TM misalignments. This means that the SM is more capable of compensating for wave aberrations, that is, the perturbation errors (the TM misalignments and the PM figure errors) that can be tolerated is greater.

## 5. Conclusions

Based on NAT, this paper studies the wave aberration compensation correction method of an unobscured off-axis TMA astronomical telescope. Firstly, third-order net astigmatism (C5/C6), third-order net coma (C7/C8), and third-order net spherical aberration (C9) induced by different perturbation parameters (misalignments and PM figure errors) in the unobscured off-axis TMA telescope are expressed analytically according to different field dependence. Furthermore, from the perspective of the compensation mechanism of aberration fields, the reason and feasibility that SM misalignments, TM misalignments, and PM figure errors can compensate each other are analyzed, and the aberration field compensation relationship shown in Equations (11)–(13) is constructed. It is worth noting that the misalignments include axial and lateral misalignments, and PM figure errors consist of third-order astigmatic figure errors, third-order coma figure errors, and third-order spherical aberration figure errors. On this basis, two aberration compensation correction models are constructed in this paper, one is to compensate for TM misalignments and PM figure errors simultaneously by adjusting SM, and the other is to compensate for SM misalignments and PM figure errors simultaneously by adjusting TM. Finally, the correctness of the two correction models was verified through a specific case simulation and Monte Carlo analyses. Our work not only contributes to a deep understanding of the coupling effect and compensation relationship of net aberration fields induced by different perturbation parameters but also provides a reference for an active optical compensation strategy of off-axis reflective telescopes.

**Author Contributions:** Conceptualization, J.W. and X.H.; methodology, J. W. and X.H.; software, X. H. and X.Z.; validation, J.W., X.H. and M.M.; formal analysis, J.W.; data curation, X.Z. and Z.C.; writing—original draft preparation, J.W.; writing—review and editing, X.H. and X.Z.; funding acquisition, X.H. and Z.C; supervision, X.H.; project administration, X.Z.; visualization, J.W.; resources, Z.C. All authors have read and agreed to the published version of the manuscript.

**Funding:** This research was funded by the National Natural Science Foundation of China (Grant No. 61875190 and 12003033), and the Natural Science Foundation of Jilin Province (Grant No. 20200201008JC).

**Institutional Review Board Statement:** Not applicable.

**Informed Consent Statement:** Not applicable.

**Data Availability Statement:** Data underlying the results presented in this paper are not publicly available at this time but may be obtained from the authors upon reasonable request.

**Acknowledgments:** The authors would like to express appreciation to the editors and reviewers for their valuable comments and suggestions.

**Conflicts of Interest:** The authors declare no conflict of interest.

## Appendix A

The optical structure parameters of the off-axis TMA telescope are shown in Table A1. The aberration coefficients for the spherical base curve and the aspheric departure from the spherical base curve of each surface are shown in Table A2, which are directly obtained from the optical simulation software.

**Table A1.** Optical structure parameters of the off-axis TMA telescope.

| Surface | Radius (mm) | Thickness (mm) | Conic Constant | Decenter X (mm) | Decenter Y (mm) | Tilt About X (deg) | Tilt About Y (deg) |
|---|---|---|---|---|---|---|---|
| Object | Infinity | Infinity | 0 | – | – | – | – |
| Stop | Infinity | 0 | 0 | – | −460 | – | – |
| PM | −3600.41 | −1551.77 | −0.921 | 0 | 0 | 0 | 0 |
| SM | −910.903 | 1558.771 | −4.828 | 0 | −8.241 | 1 | 0 |
| TM | −1219.413 | −1533.43 | −0.292 | 0 | −24.122 | 1.752 | 0 |
| Image | Infinity | 0 | – | – | – | – | – |

**Table A2.** Aberration coefficients for each individual surface ($\lambda$ = 632.8 nm).

| Surface | | $W_{222}$ | $W_{131}$ | $W_{040}$ |
|---|---|---|---|---|
| PM(stop) | Base sphere | 10.4480 | −37.1738 | 33.0656 |
| | Aspheric departure | 0 | 0 | −30.4534 |
| SM | Base sphere | −5.7486 | 11.0041 | −5.2660 |
| | Aspheric departure | 11.0197 | 12.5543 | 3.5756 |
| TM | Base sphere | 12.5832 | 2.7675 | 0.1521 |
| | Aspheric departure | −28.0608 | 10.9087 | −1.0601 |

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
