# Peer review of "Wave Aberration Correction for an Unobscured Off-Axis Three-Mirror Astronomical Telescope Using an Aberration Field Compensation Mechanism"

_applsci, doi:10.3390/app122110716_

Round 1

Reviewer 1 Report

The manuscript entitles "wave aberration correction for an unobserved off axis three mirror astronomical telescope using aberration field compensation mechanism" by jinxin wang et al. reports about corrections to specific optic issues which can be found in complex telescopes.

The research has been conducted with care and considering all the major aspects of the aberration issues when combining multiple mirrors.

Despite the conclusions are very specific and therefore the scientific interest resonates only with a small number of research groups, the presented work is worth publication on "Applied Sciences".

Reviewer 2 Report

The author presents the high-quality work on aberration correction solution for three mirror structure telescope. While focusing on the major term C5, C6, C7, C8, and C9.  This would be a very good reference for scientific community. 

(1) Please briefly explain the reason to pick 632.8 nm. 

(2) In line #210, you said "so that the different components can compensate each other". Technically, I agree with you on that point. But would you please explain that with more details. 

Reviewer 3 Report

This work theoretically studies the wave aberration compensation correction method of an unobscured off-axis TMA astronomical telescope. The  analysis of the coupling effects and compensation relationships of net astigmatism, net coma, and net spherical aberration fields induced by different perturbation parameters) of the telescope is described properly. However, the simulation part is vague and needs more details. In my judgment, the paper can be published after adding the details of the used simulation method as long as improving the language.

Reviewer 4 Report

The article is interesting. 

The originality is average because the article is focused on the principle description.  

I think the abstract is a little bit longer. 

In the final paragraph of the ”Introduction”, a short summary of the following chapters of the article was expected.

I recommend the publication of the article with small adjustments to the text.
